



# Improved CASA model based on satellite remote sensing data: Simulating net primary productivity of Qinghai Lake Basin alpine grassland

Chengyong Wu[1,3], Kelong Chen[2,3], Chongyi E[2,3], Xiaoni You[1], Dongcai He[1], Liangbai Hu[1], Baokang Liu[1], Runke Wang[1], Yaya Shi[1], Chengxiu Li[1], and Fumei Liu[2]

[1]School of Resources and Environmental Engineering, Tianshui Normal University, Tianshui, 741001, China
[2]MOE Key Laboratory of Tibetan Plateau Land Surface Processes and Ecological Conservation / Qinghai Province Key Laboratory of Physical Geography and Environmental Processes, Xining, 810008, China
[3]Academy of Plateau Science and Sustainability, Xining, 810008, China

*Correspondence to*: Chengyong Wu (giswuchengyong@163.com)

**Abstract.** The Carnegie-Ames-Stanford Approach (CASA) model is widely used to estimate vegetation net primary productivity (NPP) at regional scale. However, the CASA is still driven by multi-source data, e.g. satellite remote sensing (RS) data, and ground observations that are time-consuming to obtain. RS data, can conveniently provide real-time surface information at the regional scale, thus replacing ground observation data to drive CASA model. We attempted to improve the CASA model in this study using DEM data derived from radar RS and RS products data generated from Moderate Resolution Imaging Spectroradiometer satellite sensor. We applied it to simulate the NPP of alpine grasslands in Qinghai Lake Basin, which is located in the northeastern Qinghai-Tibetan Plateau, China. The accuracy of the RS data driven CASA, with mean absolute percent error (MAPE) of 23.32% and root mean square error(RMSE) of 26.26 g C•m$^{-2}$•month$^{-1}$, was higher than that of the multi-source data driven CASA, with MAPE of 49.08% and RMSE of 65.21 g C•m$^{-2}$•month$^{-1}$. The NPP simulated by RS data driven CASA in July 2020 shows an average value of 110.17±26.25 g C•m$^{-2}$•month$^{-1}$, which is similar to published results and comparable with the measured NPP. The results of this work indicate that simulating alpine grassland NPP with satellite RS data rather than ground observations is feasible. We may provide a workable reference for rapidly simulating grassland, farmland, forest, and other vegetation NPP to satisfy the requirements of precision agriculture, precision livestock farming, accounting carbon stocks, and other applications.

## 1 Introduction

Net primary productivity (NPP) is defined as the net accumulation of organic matter through photosynthesis by green plants per unit of time and space (Yu et al., 2009). NPP reflects the carbon sink, production, and food supply capacity of an ecosystem (Jiao et al., 2018; Li et al., 2019), so it plays an important role in studying carbon cycles, ecosystem management, grassland productivity (Zhang et al., 2016), crop yields (Wang et al., 2019), climate change (Zhang et al., 2018), and other issues directly or indirectly at both local and global scales (Li et al., 2020). NPP has been the subject of a great deal of attention from academics and governmental agencies (Wang et al., 2017). It is also a necessary input parameter for many





models in the research of global change and ecology. Accordingly, it has been recognized by the International Biological Program (IBP) (Uchijima and Seino, 1985), the International Geosphere Biosphere Program (IGBP Terrestrial Carbon Working Group, 1998), Global Change and Terrestrial Ecosystem (GCTE) (Fang et al., 2003), and the Kyoto Protocol as a
key indicator.

Direct field measurements are time-consuming and costly, so simulation models are generally used to analyze NPP (Hadian et al., 2019). Existing NPP simulation models can be roughly split into three categories: climate relative models, process models, and Light Use Efficiency (LUE) models. LUE models include the Carnegie-Ames-Stanford Approach (CASA) model (Potter et al., 1993; Field, et al., 1995), carbon fixation model (Veroustraete et al., 2002), and carbon flux model
(Turner et al., 2006). Among them, the CASA is a mechanistic model that describes processes of carbon exchange between the terrestrial biosphere and atmosphere (Cramer et al., 1999); it has been widely used to simulate regional or continental NPP over hundreds of published studies (Jay et al., 2016).

The parameters of CASA model are total solar radiation (SOL), water stress coefficient (WSC), fraction of absorbed photosynthetically active radiation (FPAR), temperature stress factors $T_{\varepsilon 1}$ (the temperature at which the plant can perform
photosynthetic activities) and $T_{\varepsilon 2}$ (the temperature at which the plant can efficiently use the light), and the maximum possible efficiency ($\varepsilon_{max}$). At present, the FPAR and $\varepsilon_{max}$ have been driven by remote sensing (RS) data. $T_{\varepsilon 1}$ and $T_{\varepsilon 2}$ are usually calculated by the air temperature data from ground meteorological stations through spatial interpolation method. SOL, as basic driver of CASA model, due to lack of measured data, were usually calculated by Angstrom-Prescott equation, or were estimated by solar radiation flux (SolarFlux) model. The Angstrom-Prescott equation (Prescott, 1940) uses measured solar
radiation data to determine its empirical coefficients a(the ratio of surface solar radiation to astronomical radiation under the completely cloudy condition) and empirical coefficients b(reflecting the transmission characteristics of clouds to solar radiation), and then calculates SOL using sunshine duration data from ground meteorological station. The empirical coefficients a and b will change as the time and territories change. In addition, this method lacks a meteorological basis that weather conditions such as cloudy sky or clear sky are determined by the total cloud cover, are not depended on the number
of sunshine duration. The SolarFlux model, can simulate SOL, and its simulation precision mainly depends on the accuracy of atmospheric conditions. WSC, as another basic driver of CASA model, meaning the availability of water, in traditional studies, was obtained using a ratio of the actual\estimated evapotranspiration (ET) to the potential evapotranspiration (PET). Initially, both ET and PET came from soil moisture submodel. This model need the meteorological data of temperature and precipitation, and soil texture, soil depth, and other soil parameters usually obtained from soil database or field investigation.
As the study progressed, ET and PET were calculated separately with different simulation model and data source. Usually, PET can be calculated by FAO Penman-Monteith equation (Allen et al., 1998) that needs meteorological observation data such as minimum temperature, maximum temperature, air temperature, wind speed, relative humidity and sunshine duration. ET can be obtained with models based on complementary relationship of evapotranspiration (Bouchet, 1963) or other approaches such as Pike equation (Pike, 1964). In view of numerous parameters, difficulty in obtaining, and complicated
calculation, most scholars have improved WSC through modifying the calculation of ET or PET (e.g., Xu and Wang, 2016;





Zhang et al., 2016; Pei et al., 2018). A few scholars attempted to introduce RS data for improving WSC, but still need the support of ground observation data, e.g., Bao et al. (2016) introduced RS data and proposed the land-surface water index and the ScaledP (the ratio between monthly precipitation amounts and the maximum monthly precipitation for individual pixels of precipitation) to improve WSC.

In summary, some parameters of the CASA model were still obtained from meteorological data, measured solar radiation, soil data and other ground observation points data. Usually, the spatial distribution of these ground observation points are few and scattered, especially in a small region, there may be only a few or even no observation stations, which affects the application of CASA model. Moreover, due to the CASA need to input continuous raster data, it means that the data of discrete observation points must be convert into continuous raster data of study area, which inevitably takes errors, and in

turn affects the accuracy of simulation NPP. In addition, soil field measurements are time-consuming, and the monthly meteorological data and measured solar radiation data from meteorological departments often were published in time-delay, which makes it impossible to estimate NPP in real time, and cannot meet the application requirements of precision agriculture, precision livestock farming, accounting carbon stocks, etc. Hence the CASA model driven by multi-source data such as meteorology, soil, and RS has notable disadvantages. Compared to these ground observation points data, satellite RS

can rapidly obtain land surface data at regional scale. Moreover, with the development of satellite sensors and RS algorithms, many quality-controlled RS products have been produced and are available online. Therefore, we hope to use entire RS data to drive CASA model. To achieve this, using the Moderate Resolution Imaging Spectroradiometer (MODIS) RS products and Digital Elevation Model (DEM) data derived from radar RS, we attempts to  improve CASA model and its parameters as follows: (1)SOL was driven by cloud cover data from MOD08_M3 product and the DEM data; (2)$T_{\varepsilon 1}$ and $T_{\varepsilon 2}$ were driven by

land surface temperature data from MOD11A2 product; (3)SWC was driven by shortwave infrared reflectance data from MOD09A1 product; (4)FPAR was driven by Normalized Difference Vegetation Index (NDVI) data from MOD13A1 product; and (5)the RS data driven CASA model was tested with multi-source data driven CASA model and the measured NPP of alpine grassland in Qinghai Lake Basin, in the northeast of Qinghai Tibet Plateau, China.

## 2 Data sources

### 2.1 Study area

Qinghai Lake Basin is located in the northeastern part of the Qinghai-Tibetan Plateau (Fig. 1). Its topography varies greatly over an altitude range of 3193-5114 m. It has a cold climate with an average annual air temperature of 1.2 ℃ (1951-2007). Its main vegetation types are alpine grasslands and alpine meadows, which account for 85.31% of all vegetation types. Qinghai Lake Basin was taken here as a typical empirical study area to test the proposed RS-driven CASA model under conditions of

varied topography and relative single vegetation types.



### 2.2 Data sources

#### 2.2.1 DEM

DEM data with 90 m spatial resolution was derived from the Shuttle Radar Topography Mission as provided by the Geospatial Data Cloud (http://www.gscloud.cn/). It was aggregated into 500 m spatial resolution on the ARCGIS 10 software platform, then used to calculate SOL.

#### 2.2.2 Solar radiation measurements

There is only one provincial ground solar radiation observation station in the study area. Observation data for the station in 2020 were not yet published at the time of this study, so we obtained its monthly SOL data for 2005, 2010, and 2015 from China Meteorological Data Service Center (http://data.cma.cn/) to verify the SOL simulation.

#### 2.2.3 Ground meteorological data

The meteorological data of twenty ground observation stations in the study area and surrounding areas,were obtained from China Meteorological Data Service Center (http://data.cma.cn/) and Qinghai Climate Center, Qinghai Province,China. These average monthly data for 2005,2010, 2015, and 2020, including temperature (mean, minimum,maximum), sunshine duration (only for 2020), sunshine percentage, precipitation, wind speed, and relative humidity, were used to calculate traditional SOL and WSC, and as input parameter of multi-source data driven CASA model.

#### 2.2.4 Land-use and Land-cover change data

Land-use and land-cover change data with 30 m spatial resolution in 2020, as a Geo-information Public Product, were obtained from GlobeLand30 (http://www.globallandcover.com/) to identify grassland types.

#### 2.2.5 RS data

MODIS is a key sensor aboard the Terra and Aqua satellites. Terra MODIS and Aqua MODIS are covering the entire earth's surface every one to two days. The Earth Science Data Systems Program generates 8-day, 16-day, monthly, and other time-scaled quality-controlled MODIS products. The products MOD11A2, MOD09A1, MOD13Q1, and MOD08M3 were obtained from the National Aeronautics and Space Administration (NASA, https://ladsweb.modaps.eosdis.nasa.gov/search/). MOD 13Q1, MOD 09A1, and MOD 11A2, with spatial resolution ranging from 250 m to 1000 m, were resampled to 500 m spatial resolution via bilinear interpolation method, then used to calculate CASA model parameters. MOD08M3 was used to count total cloud cover without necessarily adjusting its spatial resolution.

AMSR2 products, a surface soil moisture data set, have been evaluated in several previous studies, and compared quite well with both observational and model simulation data sets from a variety of global test sites (Owe et al., 2008).We obtained the



daily LPRM_AMSR2_DS_A_SOILM3 data of AMSR2 products in July 2020 from the Goddard Distributed Active Archive Center (DAAC, https://disc.gsfc.nasa.gov/), and averaged together to evaluate the simulation results of WSC.

### 2.2.6 Field observation data

The field observation NPP data were surveyed via quadrat method. Referencing the Technical regulations for Survey and Collection Biomass of Forest Carbon Pools (SACINFO, 2021) and other approaches of ground survey of grass NPP, we designed three 1 m × 1 m quadrats in the corner of square sample plots 25 m × 25 m in size. The average NPP values of
these three quadrats was regarded as the NPP value of the sample plot. All vegetation above ground in the quadrat was cut with scissors and placed into self-sealing bags, then placed into an oven at 105°C, baked for 15 min, and dried at 65 °C until a constant dry biomass value. The dry aboveground biomass (AGB) value was converted to NPP as follows (Zhang, 2016):

$$NPP = AGB \times C(1 + SR),  \tag{1}$$

where $C$ is carbon content coefficient converting biomass to NPP. It does not exceed 40% for herbaceous plants in the Tree-
River Headwaters Region, Qinghai-Tibetan Plateau (Sun et al., 2017), and was set to 37.13% here according to the average carbon content of herbaceous plants (Zheng et al., 2007). $SR$ represents the ratio of above-ground biomass to below-ground biomass. Liu et al. (2020) reported that the average root-shoot ratio (the ratio of below-ground and above-ground biomass) of alpine grassland is 6.87, so $SR$ was set to 1.00/6.87, namely $SR$ equals 0.146 in this case.

From July 23 to July 27, 2020, we investigated a total of 30 quadrats and obtained ten samples of NPP data to validate the
RS-driven CASA model (Table 4).

## 3 Methods

### 3.1 CASA model

The CASA model incorporates meteorology, environment, and soil factors to simulate the physiological process of vegetation absorbing photosynthetically available radiation and transforming it into organic carbon. The model is given as
follows (Potter et al., 1993; Wang et al., 2017):

$$NPP(x,t) = 0.5 \times SOL(x,t) \times FPAR(x,t) \times T_{\varepsilon 1} \times T_{\varepsilon 2} \times WSC(x,t) \times \varepsilon_{max},  \tag{2}$$

where $NPP$ is the net primary production (g C•m$^{-2}$•month$^{-1}$); 0.5 represents the proportion of the radiation which can absorbed by plants; $SOL$(x,t) is the total solar radiation incident on grid cell x in a given month (MJ•m$^{-2}$•month$^{-1}$); $FPAR$(x,t) is the fraction of absorbed photosynthetically active radiation on grid cell x in a month; $T_{\varepsilon 1}$ and $T_{\varepsilon 2}$, the temperature stress factors, account for the effect of high and low temperature on light utilization efficiency, respectively; $WSC$(x,t) is the water
stress coefficient on grid cell x in a month; and $\varepsilon_{max}$ is the maximum possible efficiency (g C•MJ$^{-1}$) under ideal conditions (no-stress temperature, no-stress water).





### 3.2 Improving CASA parameters with RS data

The RS data utilized here to improve CASA parameters are listed in Table 1. We focused specifically on improving the
parameters SOL and WSC.

### 3.2.1 Calculation SOL by introducing RS cloud cover

SolarFlux models (Hetrick et al., 1993; Kumar et al., 1997; Fu and Rich, 2002), which input DEM parameters and compute solar radiation over large areas, have been implemented for commercially available GIS software such as ARC/INFO, ARCGIS, and Genasys. The solar radiation module of ARCGIS software takes into account the influence of atmospheric
conditions, latitude, altitude, solar zenith angle and azimuth angle, terrain shade, slope, and aspect. The atmospheric conditions relevant to the present study were determined by the parameters diffuse_proportion and transmittivity. The diffuse_proportion is the fraction of global normal radiation flux that is diffused, which is expressed as a values from 0 to 1. Transmittivity, the fraction of radiation that passes through the atmosphere, ranges from 0 (no transmission) to 1 (all transmission) (ESRI, 2021).
There are distinct differences between diffuse_proportion and transmittivity on clear and cloudy days (i.e., dependent on total cloud cover). The accurate determination of atmospheric conditions is the key to accurately estimating SOL. We introduced satellite total cloud cover to classify weather conditions, then determined the corresponding diffuse_proportion and transmittivity values. The total cloud cover data from the MOD08_M3 product, ranging from 0 (where the sky is completely clear) to 10,000 (where the sky is completely covered by clouds), was divided by 1,000 to create ten levels. For
each level, the diffuse_proportion and transmittivity were determined according to a simple linear relationship (Table 2).

### 3.2.2 Improvement WSC using shortwave infrared reflectance

WSC reflects the effect of available water content on the solar radiation utilization efficiency of plants, ranging from 0.5 (extreme drought conditions) to 1.0 (extreme humidity). According to the RS principle that shortwave infrared reflectance is negatively correlated with surface water content, scholars have proposed many water content RS indices. Referring to the
form and connotation of the shortwave infrared soil moisture index (SIMI) proposed by Yao et al. (2011), we rewrote the WSC formula as follows:

$$WSC = 0.5 + 0.5(1 - N_{SIMI}) ,  \tag{3}$$

$$N_{SIMI} = (SIMI - SIMI_{min})/(SIMI_{max} - SIMI_{min}) ,  \tag{4}$$

$$SIMI = 0.7071\sqrt{SWIR_1^2 + SWIR_2^2} ,  \tag{5}$$

where $WSC$ is the water stress coefficient; $N_{SIMI}$ represents the normalized SIMI, which values range from 0 to 1; $SIMI_{max}$ and $SIMI_{min}$ are the maximum and minimum value of $SIMI$ values, respectively; $SWIR_1$ and $SWIR_2$ are the shortwave infrared reflectance values of band 6 and band 7 from MOD09A1, respectively.





## 4 Results

### 4.1 SOL

#### 4.1.1 SOL simulated by Angstrom-Prescott equation

Angstrom-Prescott equation, a traditional approach for simulation SOL, was used to calculate the SOL of ground meteorological stations. Its empirical coefficients a and b were adopted the monthly coefficients of Liu et al. (2021), and its input parameters S (sunshine percentage) from ground meteorological stations. Natural Neighbor spatial interpolation approach was applied to convert the SOL of ground stations into grid WSC over study area (Fig. 2-A).

#### 4.1.2 SOL simulated by improved approach

The DEM, diffuse_proportion, and transmittivity determined by MODIS total cloud cover were input into the Solar Radiation module of ARCGIS10 software. The SOL in July of 2020 was simulated in Qinghai Lake Basin (Fig. 2-B) ranging from 655.42 MJ•m$^{-2}$•month$^{-1}$ to 878.03 MJ•m$^{-2}$•month$^{-1}$ with an average value of 738.80 MJ•m$^{-2}$•month$^{-1}$.The surface of Qinghai Lake shows the lowest SOL, 695.50 MJ•m$^{-2}$•month$^{-1}$. On the whole, SOL gradually increases along Qinghai Lake 195 from southeast to northwest and are basically consistent with the actual total solar radiation in Qinghai Lake Basin**.**

#### 4.1.3 Comparison of two SOL simulation approaches

We analysed the accuracy of simulation SOL from Angstrom-Prescott equation and improved SOL approach with the measured SOL monthly data in 2005, 2010, and 2015 (at present, only the measured SOL data in these period could be collected for the purposes of this study). We simulated SOL in the same period and analysed its accuracy accordingly (Table 200 3). The root mean square error (RMSE) of Angstrom-Prescott equation and our improved approach respectively are 162.24 MJ•m$^{-2}$•month$^{-1}$ and 95.38 MJ•m$^{-2}$•month$^{-1}$.Correspondingly, the mean absolute percent error (MAPE) of two approaches are 24.56% and17.78%, the July RSME are 274.34 MJ•m$^{-2}$•month$^{-1}$ and 70.66 MJ•m$^{-2}$•month$^{-1}$, and the July MAPE are 39.53% and 9.25%, respectively. Obviously, for simulating SOL in study area, our improved approach is superior to Angstrom-Prescott equation.

### 4.2 WSC

#### 4.2.1 Traditional WSC

Traditionally, the WSC was obtained using a ratio of ET to PET. Using ground meteorological data for July 2020, we applied FAO Penman-Monteith equation (Allen et al., 1998) to calculate PET and adopted Pike equation (Pike, 1964) to calculate ET, and then obtained the WSC of ground observation stations. Natural Neighbor approach was used to convert the 210 WSC of ground stations into grid WSC over study area (Fig. 3-A).



### 4.2.2 Improved WSC

Using RS shortwave infrared reflectance from product MOD09A1, We applied Eq(3), Eq(4) and Eq(5) and obtained the WSC in July, 2020(Fig. 3-C).The WSC values in July, 2020, were relatively high (>0.86) around Qinghai Lake and in river valleys as well as in the river source areas at higher altitudes indicating that the ecosystem has sufficient water supply (Fig. 3-B). The desert ecosystem in the east of the Qinghai Lake showed the lowest WSC (0.54-0.68) indicating that the ecosystem has insufficient water supply.

### 4.2.3 Comparison of two WSC simulation approaches

WSC, measuring the availability of water by plants, in essence, reflects the impact of environmental water content on plants. For grassland ecosystem, to a certain extent, surface soil moisture (SM) can indirectly reflect environmental water content. As a general rule, a higher value of WSC indicates a higher environmental water content. We use the surface SM data set (LPRM_AMSR2_DS_A_SOILM3, as mentioned in section 2.2.5, its accuracy have been tested in several previous studies) to evaluate the WSC results simulated by different approaches.

The SM is high in north of Qinghai Lake (Region N), and it is the lowest in the desert ecosystem (Fig. 3-B).

In region N, the traditional WSC showed low values indicating that environmental water content is low, and the desert ecosystem showed a lower values, but not the lowest. Hence the traditional WSC results are inconsistent with surface SM, it cannot reflect the spatial distribution of environmental water content. The reason is that the sparse distribution of ground meteorological stations causes the uncertainty of WSC interpolation results.

The improved WSC simulation results compared well with the surface SM in above two regions, its spatial distribution are approximately consistent with the actual water contents in study area, so it is feasible to estimate WSC using RS shortwave infrared reflectance.

### 4.3 NPP

### 4.3.1 Comparison of multi-source and RS data driven CASA

We used the measured NPP obtained in July of 2020 to verify the accuracy of multi-source and RS data driven CASA model (Table 4). For multi-source data driven CASA (Fig. 4-A), its parameters SOL, SWC, $T_{\varepsilon 1}$, and $T_{\varepsilon 2}$ come from ground meteorological data, and the FPAR and $\varepsilon_{max}$ are as same as the parameters of RS data driven CASA, the relative error (RE) ranges from 30.98% to 85.88%, the MAPE is 49.08%, the absolute error (AE) ranges from 24.55 g C•m$^{-2}$ •month$^{-1}$ to 141.66 g C•m$^{-2}$•month$^{-1}$, and the RMSE is 65.21 g C•m$^{-2}$•month$^{-1}$.For NPP simulated by RS data CASA, the RE ranges from 5.66% to 50.02%, the MAPE is 23.32%, the AE ranges from -49.08 g C•m$^{-2}$•month$^{-1}$ to 23.89 g C•m$^{-2}$•month$^{-1}$, and the RMSE is 26.26 g C•m$^{-2}$•month$^{-1}$. The simulation results of RS data driven CASA are more in accordance with the measured NPP, RS data driven CASA is superior to the multi-source data driven CASA.





### 4.3.2 NPP spatial distribution

The NPP values in July, 2020, are lower in the northwest parts of the basin and east of Qinghai Lake than elsewhere in the study area (Fig. 4). The main vegetation in the northwest is Alpine Kobresia humilis meadow plants such as *Saussurea pumila* and *Saussurea alpina*, which have low vegetation productivity and NPP values ranging from 1.09 g C•m$^{-2}$•month$^{-1}$ to
87.85 g C•m$^{-2}$•month$^{-1}$. The main vegetation in the southwest coast of Qinghai Lake and the middle part of the basin are *Stipa purpurea Griseb* and *Carex infuscata Nees* alpine grasslands, which have higher vegetation productivity and NPP values greater than 87.85 g C•m$^{-2}$•month$^{-1}$. NPP appears to decrease from southeast to northwest, which is consistent with the distribution patterns of vegetation type.

## 5 Discussion and recommendations

### 5.1 SOL

When astronomical solar radiation passes through the atmosphere, it is weakened by atmospheric scattering and absorption, and finally transmits to earth surface (so called surface solar radiation),which means that atmospheric conditions significantly affect surface solar radiation. Various approaches for simulation SOL consider the atmospheric effects on solar radiation from different perspectives. Angstrom-Prescott equation uses the sunshine duration (or sunshine percentage) to
quantify atmospheric effects on solar radiation.We use the parameters of diffuse_proportion and transmittivity determined by total cloud cover to quantify this effects. The total cloud cover determines the weather conditions, it also affects the atmospheric conditions. Total cloud cover information can be used to directly determine weather conditions and indirectly determine atmospheric conditions. In this study, weather conditions were classified into ten levels according to the satellite total cloud cover. The two important parameters of the SolarFlux model, diffuse_proportion and transmittivity, were
determined for each level on the basis of a linear relationship. The atmospheric conditions could be further divided into 100 or more refined levels to determine the values of diffuse_proportion and transmittivity under different cloud cover conditions to improve the SOL simulation accuracy.

It is important to note that the SolarFlux model is designed only for local landscapes\regional scales, so it is generally acceptable to use one latitude value for the whole DEM. It is necessary to divide larger areas into zones of varying latitude as
the latitudes exceed 1 degree (ESRI, 2021).

### 5.2 WSC

Environmental water content can regulate vegetation NPP by affecting the photosynthetic capacity of plants. WSC reflects the influence of environmental water content on vegetation NPP. Traditional WSC simulation approach apply a ratio of ET to PET to measure the availability of environmental water content. ET and PET were obtained by different approaches and
data sources. It means that there are great differences in ET and PET, even if the same data is used, which result in





differences in WSC. The WSC result of our improved approach is unique, as long as the same RS data is input in formula (3), (4), and (5). In addition, our improved WSC approach has the RS retrieval mechanism of environmental water content. Soil and vegetation water contents are closely related to their shortwave infrared spectral reflectance; small changes in these contents can cause substantial changes in shortwave infrared spectral reflectance. Thus, the RS shortwave infrared band is

sensitive to environmental water content and can be used to calculate WSC. Many satellite sensors are designed with shortwave infrared bands that are extremely sensitive to water content, such as MODIS (1.628-1.652 μm, 2.105-2.155 μm), LandSat 8 (1.560-1.660 μm, 2.100-2.300 μm), Sentinel-2(1.565-1.655 μm, 2.100-2.280 μm), and HJ-1-A, B (1.550-1.750 μm). Scholars have developed many RS water content indexes such as SIMI, MSIWSI (Dong et al., 2015) and SWCI (Du et al., 2007). We modified the WSC using SIMI and the two shortwave infrared bands of MODIS in this study. The shortwave

infrared bands of satellite sensors mentioned above, as well as the MSIWSI, SWCI, or other RS water content indices, can also be considered to calculate WSC.

### 5.3 Temperature stress factors

Excessively low temperature can limit plant photosynthesis and excessively high temperature can increase the respiration consumption of plants. If the temperature is not optimal, the LUE of vegetation decreases thereby reducing its NPP.

Temperature stress factors $T_{\varepsilon1}$ and $T_{\varepsilon2}$ represent the effects of low temperature and high temperature on LUE. Previous researchers have mainly used air temperature data from ground meteorological observation station to calculate these factors via interpolation method. Land surface temperature (LST) products show strong accuracy, e.g., MODIS LST accuracy over 1 K under clear sky conditions (Wan et al., 2002). With the development of RS sensors, LST products have been gradually enriched. We used the MODIS 8-day LST product in this study to drive $T_{\varepsilon1}$ and $T_{\varepsilon2}$; other LST products (e.g., ASTER, FY-3,

Landsat series, and Sentinel-3) can be selected according to the characteristics of the study area and study period.

### 5.4 Rationality of RS data driven CASA

### 5.4.1 Rationality of simulation results

We compared our simulated NPP with previously published results (Table 5). Our simulated grassland NPP in July of 2020 has an average value of 110.17 ±26.25 g C•m$^{-2}$•month$^{-1}$, which is similar to the most published results, but smaller than some

of them. Qinghai Lake Basin is located on the Qinghai-Tibetan Plateau, which has a severely cold climate and short growing season. Vegetation is in its growth stage in July, where biomass reaches the highest values for the whole year before declining at the end of August or beginning of September. The reported NPP encompasses the full year, so it is reasonable that July NPP simulation values would be lower than some previously reported NPP values.





### 5.4.2 Rationality of RS mechanism

In this study,the simulation NPP values of *Kobresia parva* and *Stipa purpurea* respectively are larger and smaller than the measured NPP values.

*Kobresia parva* is distributed in high-altitude areas which herdsmen often utilize as summer pastures. Grazing cattle and sheep reduce the biomass of these areas resulting in lower measured NPP values. *Kobresia parva* is characterized by low and short (1-3 cm) vegetation with densely clumped stems and high coverage. Grazing livestock does not significantly affect its

reflectance at red and near infrared bands. For grazed and ungrazed *Kobresia parva*, the NDVI calculated by the reflectance of red and near infrared bands are almost the same; the FPAR values calculated by NDVI are also very similar, so the simulated NPP values are nearly identical as well. Due to the lower measured NPP value of *Kobresia parva* caused by grazing, the NPP simulation values of *Kobresia parva* appear to be relatively high.

*Stipa purpurea*, which is distributed in low-altitude areas which herdsmen often use as winter pastures, is an ideal vegetation

type to verify the NPP model as it is not consumed by cattle, sheep, or other livestock during the summer. *Stipa purpurea* has a thin stalk up to 45 cm high and leaves curled into needles with strongly lignified epidermes and purple spikelets. These characteristics result in a lower reflectance at red and near infrared bands, which leads to lower NDVI and FPAR values. Thus, the simulated NPP values of *Stipa purpurea* are relatively low.

### 5.5 Uncertainty

According to equation (1), the uncertainty of measured NPP come from uncertainty of obtaining AGB, C, and SR. There is randomness in which three quadrats are selected from the four corners of square sample plot, resulting in the uncertainty of collection AGB. In our case, C and SR are adopted the values reported in the literatures instead of the measured values, which inevitably brings errors.

The uncertainty of multi-source data driven CASA and its parameters is mainly caused by spatial interpolation methods. For

instance,the WSC interpolation results from Spline and Kriging method showed significantly different values and spatial patterns (Fig. 5).

The uncertainty of RS data driven CASA mainly stem from RS product data quality and uncertainty propagation from parameters. RS product usually have corresponding data quality assurance describing the uncertainty of each pixel (e.g., the uncertainty of production MOD11A2, for details, please see its instruction for quality assurance at:

https://icess.eri.ucsb.edu/modis/LstUsrGuide/usrguide_index.html).The combined uncertainty of simulation NPP is determined by the uncertainty propagation from parameters. In our case, the combined uncertainty of grassland NPP is $110.17\pm26.25$ g C•m$^{-2}$•month$^{-1}$. The combined uncertainty of alpine meadow and other grassland types, and uncertainty propagation and quantification, will be carried out in future work.



## 6. Conclusions

The traditional CASA model driven by multi-source data such as meteorology, soil, and RS has notable disadvantages. In this study, we attempted to drive the CASA entirely by RS data. We conducted a case study of alpine grasslands in Qinghai Lake Basin to find that it is feasible to calculate the CASA parameters SOL, WSC, $T_{\varepsilon1}$, and $T_{\varepsilon2}$ using RS data. The estimated NPP results were reliable. The main conclusions of this work can be summarized as follows.

• Cloud cover was used to quantify the atmospheric effects on solar radiation. It's only necessary to use DEM and RS
total cloud cover data for simulating SOL. The improved SOL simulation approach (the monthly RMSE and MAPE respectively were 95.38 MJ•m$^{-2}$•month$^{-1}$ and 17.78%) is superior to Angstrom-Prescott equation (a traditional approach for simulation SOL, its monthly RMSE and MAPE respectively were 162.24 MJ•m$^{-2}$•month$^{-1}$ and 24.56%).

• According to RS retrieval mechanism of environmental water content, shortwave infrared reflectance was used to estimate the WSC. The improved WSC simulation approach conforms to the implications of original model and simplified
the input parameters, its simulation results are more approximately consistent with the actual environment water contents than that of the traditional WSC in study area.

• RS data driven CASA, without the support of ground observation data (e.g., soil or meteorology), is superior to the multi-source data driven CASA, and its simulation results are more in accordance with the measured NPP. The RE ranges from 5.66% to 50.02%, the MAPE is 23.32%, the AE ranges from -49.08 g C•m•month$^{-1}$ to 23.89 g C•m$^{-2}$•month$^{-1}$, and the
RMSE is 26.26 g C•m$^{-2}$•month$^{-1}$. The NPP simulation values of *Kobresia parva* in grazing area and *Stipa purpurea* respectively are higher than and less than its real values. The combined uncertainty of grassland NPP is 110.17 ±26.25 g C•m$^{-2}$•month$^{-1}$. The uncertainty propagation and quantification will be carried out in future work.

*Code and data availability.* The code and data are available at supplement.


*Supplement.* The supplement related to this article is available online at: https://doi.org/....../gmd......-supplement.

*Author contributions.* CW, CE, KC, XY, and DH contributed to the manuscript writing. CW contributed to the code writing. LH, BL and RW contributed to data processing. CW, YS and FL contributed to field investigation. YS, CL and FL
contributed to laboratory experiment.

*Competing interests.* The authors declare that they have no conflict of interest.

*Acknowledgements.* We gratefully acknowledge the Geospatial Data Cloud, China Meteorological Data Service Center,
GlobeLand30 and NASA for providing the data of DEM,SOL, LUCC and MODIS products.



*Financial support.* This research has been supported by the National Philosophy and Social Science Foundation of China (14XMZ072,18BJY200), the National Natural Science Foundation of China (41761017,41661023), the Project of the Qinghai Research Center of Qilian Mountain National Park(GY1908), the Fuxi Innovation Team Project of Tianshui Normal University(FX202006), the Scientific and Technological Program of Gansu Province, China (21JR1RE293), the Project of the State Key Laboratory of Frozen Soil Engineering(SKLFSE202014), and the Colleges and Universities Innovation Ability Improvement Project of Gansu Educational Committee (2019B–134).

*Review statement.*

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

![Figure 1 map showing location of Qinghai Lake Basin with altitude, sample sites and observation stations]

**Figure 1. Location of Qinghai Lake Basin, sample and ground observation points.**






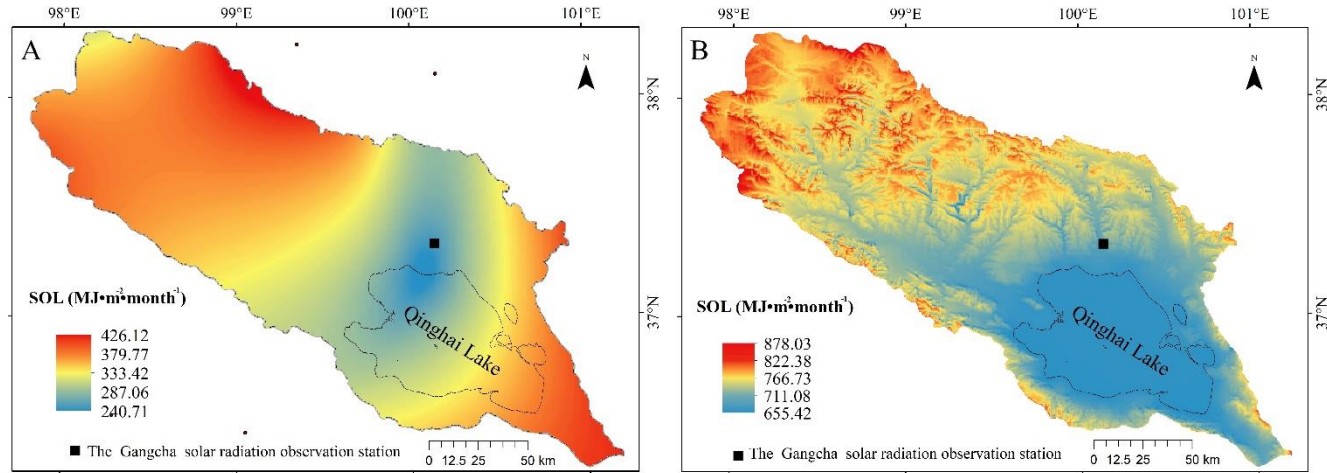

**Figure 2. Spatial distribution of total solar radiation (SOL) in July, 2020. A, SOL simulated by Angstrom-Prescott equation. B, SOL simulated by improved approach.**






**Figure 3. Spatial distribution of water stress coefficient (WSC) in July, 2020. A, WSC simulated by traditional method. B, Surface soil moisture of AMSR2 products. C, WSC calculated with RS shortwave infrared band.**

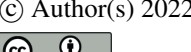



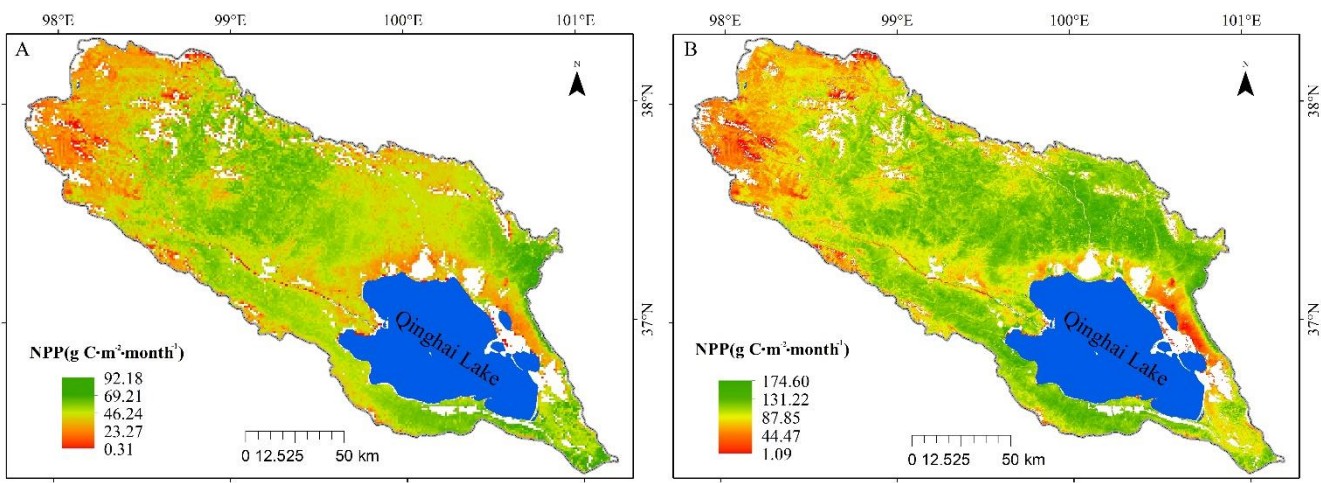

**Figure 4. Spatial distribution of grassland net primary productivity (NPP) in July, 2020. A, NPP simulated by multi-source data driven CASA. B, NPP simulated by RS data driven CASA.**

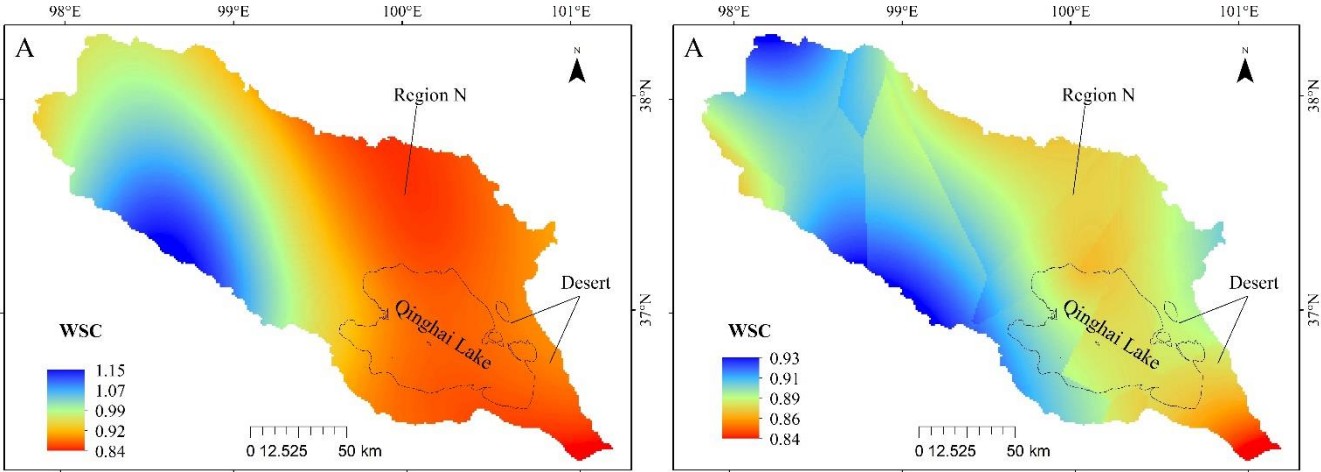

**Figure 5. Comparison map of water stress coefficient (WSC) interpolation results in July, 2020. A, WSC from Spline method. B, WSC from Kriging method.**





**Table 1. Input data and calculation method for RS-driven CASA model parameters**


| Parameter | Input data\calculation method |
|---|---|
| SOL | DEM and MOD08M3 |
| WSC | Band 6 (1.628-1.652 µm) and band 7 (2.105-2.155 µm) from MOD09A1 |
| $T_{\varepsilon 1}$, $T_{\varepsilon 2}$ | Temperature (T) and optimum temperature ($T_{opt}$) are necessary for calculating $T_{\varepsilon 1}$ and $T_{\varepsilon 2}$; MOD11A2 provides day temperature ($T_{day}$) and night temperature ($T_{night}$); T is calculated as T=0.5 ($T_{day}$+ $T_{night}$); $T_{opt}$ is the average value of T in July. The equations of $T_{\varepsilon 1}$ and $T_{\varepsilon 2}$ can be found in Potter et al. 1993. |
| $\varepsilon_{max}$ | $\varepsilon_{max}$= 0.608g C•MJ$^{-1}$, maximum possible efficiency of grassland (Running et al., 2000) |
| FPAR | Myneni and Williams (1994) found a linear relationship between FPAR and NDVI. NDVI from MOD13A1 is used to calculate FPAR (Wang et al., 2017) |

**Table 2. Diffuse_ proportion and transmittivity values under different total cloud cover levels**

| MODIS total cloud cover level | Weather conditions | Diffuse proportion | Transmittivity |
|---|---|---|---|
| 0 | Very clear sky conditions (no clouds) | 0.2 | 0.6 |
| 1 | Cloud cover accounts for 1/9 of the whole sky | 0.255 | 0.545 |
| 2 | Cloud cover accounts for 2/9 of the whole sky | 0.31 | 0.49 |
| 3 | Cloud cover accounts for 3/9 of the whole sky | 0.365 | 0.435 |
| 4 | Cloud cover accounts for 4/9 of the whole sky | 0.42 | 0.38 |
| 5 | Cloud cover accounts for 5/9 of the whole sky | 0.475 | 0.325 |
| 6 | Cloud cover accounts for 6/9 of the whole sky | 0.53 | 0.27 |
| 7 | Cloud cover accounts for 7/9 of the whole sky | 0.585 | 0.215 |
| 8 | Cloud cover accounts for 8/9 of the whole sky | 0.64 | 0.16 |
| 9 | Sky is completely covered by clouds | 0.695 | 0.105 |


According to the scientific rule that diffuse_proportion has an inverse relation with transmittivity, the diffuse_proportion and transmittivity values were set to 0.2 and 0.6, respectively, in the case of a very clear sky conditions. Under other cloud cover conditions, their values were determined according to a simple linear relationship.




**Table 3. Measured versus simulated SOL**

| Date | Measured SOL (MJ•m$^{-2}$•month$^{-1}$) | Simulated SOL (MJ•m$^{-2}$•month$^{-1}$) | Absolute error (AE) (MJ•m$^{-2}$•month$^{-1}$) | Relative error (RE) (%) |
|---|---|---|---|---|
| Jan-05 | 374.19 | 240.95(477.62) | 133.24 (-103.43) | 35.61 (-27.64) |
| Feb-05 | 427.29 | 319.23(469.44) | 108.06 (-42.15) | 25.29 (-9.86) |
| Mar-05 | 573.16 | 489.16(528.34) | 84.00 (44.82) | 14.66 (7.82) |
| Apr-05 | 638.45 | 634.05(465.35) | 4.40(173.10) | 0.69 (27.11) |
| May-05 | 736.19 | 731.24(449.60) | 4.95 (286.59) | 0.67 (38.93) |
| Jun-05 | 663.70 | 742.68(394.28) | -78.98 (269.42) | -11.90 (40.59) |
| Jul-05 | 626.92 | 710.94(385.94) | -84.02 (240.98) | -13.40 (38.44) |
| Aug-05 | 603.86 | 623.86(423.19) | -20.00 (180.67) | -3.31 (29.92) |
| Sep-05 | 493.09 | 500.53(407.90) | -7.44 (85.19) | -1.51 (17.28) |
| Oct-05 | 486.07 | 378.72(521.19) | 107.35 (-35.12) | 22.09 (-7.22) |
| Nov-05 | 398.73 | 257.36(481.56) | 141.37 (-82.83) | 35.46 (-20.77) |
| Dec-05 | 353.71 | 197.43(456.82) | 156.28 (-103.11) | 44.18 (-29.15) |
| SOL in 2005 | 6375.36 | 5826.15(5461.24) | 549.21 (914.12) | 8.61 (14.34) |
| Jan-10 | 354.87 | 262.42(484.86) | 92.45 (-129.99) | 26.05 (-36.63) |
| Feb-10 | 409.77 | 295.56(457.35) | 114.21 (-47.58) | 27.87 (-11.61) |
| Mar-10 | 555.98 | 456.14(509.99) | 99.84 (45.99) | 17.96(8.27) |
| Apr-10 | 647.71 | 634.05(496.56) | 13.66(151.15) | 2.11 (23.34) |
| May-10 | 705.07 | 731.24(449.60) | -26.17 (255.47) | -3.71 (36.23) |
| Jun-10 | 616.64 | 649.32(368.04) | -32.68 (248.60) | -5.30 (40.32) |
| Jul-10 | 741.78 | 756.37(436.54) | -14.59(305.24) | -1.97 (41.15) |
| Aug-10 | 679.30 | 705.02(443.55) | -25.72 (235.75) | -3.79 (34.71) |
| Sep-10 | 524.02 | 500.53(428.95) | 23.49 (95.07) | 4.48 (18.14) |
| Oct-10 | 496.53 | 378.72(499.47) | 117.81 (-2.94) | 23.73 (-0.59) |
| Nov-10 | 450.87 | 299.47(507.51) | 151.40 (-56.64) | 33.58 (-12.56) |
| Dec-10 | 371.24 | 181.71(446.67) | 189.53 (-75.43) | 51.05 (-20.32) |
| SOL in 2010 | 6553.78 | 5850.55(5529.07) | 703.23 (1024.71) | 10.73 (15.64) |
| Jan-15 | 383.84 | 240.95(477.62) | 142.89 (-93.78) | 37.23 (-24.43) |
| Feb-15 | 435.62 | 319.23(453.32) | 116.39 (-17.70) | 26.72 (-4.06) |
| Mar-15 | 602.04 | 489.16(509.99) | 112.88(92.05) | 18.75 (15.29) |
| Apr-15 | 677.3 | 634.05(469.81) | 43.25 (207.49) | 6.39 (30.64) |
| May-15 | 664.51 | 731.24(408.32) | -66.73(256.19) | -10.04 (38.55) |
| Jun-15 | 621.22 | 699.14(375.53) | -77.92 (245.69) | -12.54 (39.55) |
| Jul-15 | 709.44 | 797.23(432.64) | -87.79 (276.80) | -12.37 (39.02) |
| Aug-15 | 617.12 | 705.02(431.33) | -87.90 (185.79) | -14.24 (30.11) |
| Sep-15 | 483.73 | 463.64(407.90) | 20.09 (75.83) | 4.15 (15.68) |
| Oct-10 | 509.48 | 432.73(538.56) | 76.75 (-29.08) | 15.06 (-5.71) |
| Nov-15 | 370.52 | 257.36(459.33) | 113.16 (-88.81) | 30.54 (-23.97) |
| Dec-15 | 338.99 | 197.43(456.82) | 141.56 (-117.83) | 41.76 (-34.76) |
| SOL in 2015 | 6413.81 | 5967.18(5421.18) | 446.63(992.63) | 6.96 (15.48) |
| Jul-20 | / | 709.20 | / | / |

Note: The digits in parentheses "()" are the values of SOL simulated by Angstrom-Prescott equation and the correspondingly error values.




**Table 4. Measured versus simulated NPP**

| Samples | Main vegetation | Longitude | Latitude | Measured NPP (g C•m⁻²•month⁻¹) | Simulated NPP (g C•m⁻²•month⁻¹) | AE (g C•m⁻²•month⁻¹) | RE (%) |
|---|---|---|---|---|---|---|---|
| 1 | *Kobrecia parva* | 99.87586 | 37.34791 | 91.66 | 131.77 (57.40) | -40.11 (34.26) | 43.76 (37.38) |
| 2 | *Kobrecia parva* | 99.84530 | 37.37877 | 98.12 | 147.20 (63.75) | -49.08 (34.37) | 50.02 (35.03) |
| 4 | *Kobrecia parva* | 99.30971 | 37.07243 | 110.54 | 128.25 (64.92) | -17.71 (45.62) | 16.02 (41.27) |
| 6 | *Kobrecia parva* | 100.3727 | 37.42001 | 108.33 | 135.46 (52.68) | -27.13 (55.65) | 25.05 (51.37) |
| 9 | *Stipa purpurea* | 99.67833 | 37.20655 | 121.76 | 108.80 (51.45) | 12.96 (70.31) | 10.64 (57.74) |
| 8 | *Stipa purpurea* | 99.63823 | 37.17360 | 126.86 | 114.80 (59.34) | 12.06 (67.52) | 9.50 (53.22) |
| 3 | *Carex pamirensis* | 99.48503 | 37.01362 | 111.22 | 117.51 (49.44) | -6.29 (61.78) | 5.66 (55.54) |
| 10 | *Achnatherum splendens* | 100.73520 | 36.54971 | 79.25 | 101.86 (54.70) | -22.61 (24.55) | 28.53 (30.98) |
| 5 | *Achnatherum splendens* | 100.70610 | 36.93822 | 74.82 | 50.93 (43.09) | 23.89 (31.73) | 31.93 (42.41) |
| 7 | *Blysmus sinocompressus* | 99.89820 | 36.97944 | 164.95 | 145.07 (23.30) | 19.88 (141.66) | 12.05 (85.88) |
| | RMSE=26.26 g C•m⁻²•month⁻¹, MAPE=23.32% (RMSE=65.21 g C•m⁻²•month⁻¹, MAPE=49.08%) | | | | | | |

Note: The digits in parentheses "()" are the values of NPP simulated by multi-source data driven CASA and the
correspondingly error values.







**Table 5. Published versus simulated NPP**

| Vegetation type | Study area | Study period | Mean NPP (g C•m⁻²•a⁻¹) | Model\ product | Reporter |
|---|---|---|---|---|---|
| Grassland | Three-River Headwaters Region | 1988–2004 | 160.90 | GLOPEM-CEVSA | Wang et al., 2009 |
| Grassland | Three-River Headwaters Region | 2010 | 146.66 | CASA | Wo et al., 2014 |
| Grassland | Qinghai-Tibetan Plateau | 2005–2008 | 135.00 | GLO-PEM | Chen et al., 2012 |
| Grassland | Qinghai-Tibetan Plateau | 2001–2017 | 221.16 | MODIS product (MOD17A3) | Zhang et al., 2021 |
| Alpine grassland | Three-River Headwaters Region | 2004–2008 | 129.41 | CASA | Cai et al., 2013 |
| Alpine grassland | Qinghai-Tibetan Plateau | 1982–2009 | 120.80 | CASA | Zhang et al., 2014 |
| Alpine grassland | Qinghai-Tibetan Plateau | 1982–1999 | 80.00 | CASA | Piao and Fang, 2002 |
| Alpine meadow | Three-River Headwaters Region | 2004–2008 | 188.95 | CASA | Cai et al., 2013 |
| Alpine steppe | Source Regions of Yangtze and Yellow Rivers | 2000–2004 | 79.34 | MODIS product (MOD17A3) | Guo et al., 2006 |
| Alpine steppe-meadow | China | 2004–2005 | 109.03 | CASA | Wang et al., 2017 |
| Alpine meadows and tundra | China | 1982–1999 | 137.00 | CASA | Fang et al., 2003 |
| Alpine meadows and tundra | China | 1997 | 131.00 | CASA | Piao et al.,, 2001 |
| All vegetation | Source Region of Yangtze River | 2000–2014 | 100.00 | CASA | Yuan et al., 2021 |
| All vegetation | Qinghai-Tibetan Plateau | 2012–2014 | 175.10 | Biome-BGC | Sun et al., 2017 |
| All vegetation | Qinghai-Tibetan Plateau | 2012 | 208.20 | Biome-BGC | Li et al., 2020 |
| All vegetation | Qinghai-Tibetan Plateau | 1982–1999 | 125.00 | CASA | Piao et al., 2006 |
| All vegetation | Qinghai Lake Basin | 2000–2012 | 161.01 | CASA | Zhang et al., 2015 |
| All vegetation | Qinghai Lake Basin | 2001–2011 | 168.03 | CASA | Qiao and Guo, 2017 |
