# Peer review of "Improved CASA model based on satellite remote sensing data: Simulating net primary productivity of Qinghai Lake Basin alpine grassland"

_Geoscientific Model Development, 2021_

## Author Response (AR1)

Dear editors and referees,

We are grateful for all comments. Those comments are helpful for revising and improving our code and paper, especially, for the development of CASA model. We have studied the comments carefully and have made corrections to CASA model code, data and preprint. The responses to comments and the detailed corrections are as following:

**Response to Editor Comments**

**Comments**

After checking your manuscript, it has come to our attention that it does not comply with our Code and Data Policy.

https://www.geoscientific-model-development.net/policies/code\_and\_data\_policy.html

**Comment 1**

In the Code and Data Availability section of your manuscript, you state that the necessary material is available in the supplementary file provided. However, this is hardly the case. One of the main problems is that the CASA model is omitted. You include in the supplementary material several files without any structure, order or explanation about how they work, how to run them or how they are linked. Moreover, the scripts contain several paths to local disks and servers that a reader or reviewer can not access and therefore can not test.

**Response**

Dear editor, we greatly appreciate your time dealing with the preprint and assessing the review comments. According to your advice, we integrated the codes for calculating individual parameter into a CASA mode code. We also reorganized the structure of supplementary files and formed an ordered structure to better drive the CASA model.

**Changes in Code and Data**

The structure of supplementary files was reorganized and formed an ordered structure containing the folder of RS\_data\_driven\_CASA and Multi\_source\_data\_driven\_CASA. The codes were revised and formed an integrated CASA mode code.

**Comment 2**

When providing a model, we do not refer to explaining the equations used or the papers on which it is based but the actual computer code (your implementation). Therefore, we need that you provide a better description of your code, including instructions about how to run it.

**Response**

Thank you for your suggestions. To make the model code better understand, we added the essential code comments in the revised CASA model code and provided an instruction ("how to run CASA model code .PDF" file) about how to run code. Following this instruction, the revised CASA model code can automatically link driven data and get the running results.

**Changes in Code and Data**

The essential code comments were added in the revised CASA mode code. An instruction ("how to run CASA model code .PDF" file) about how to run code was provided in the revised supplement. Following this instruction, the revised CASA model code can automatically link driven data and get the running results.

In addition, the model code that provided under review (https://doi.org/10.5194/gmd-2021-258-AC1) was further optimized. This revised CASA model code can completely delete all progress files that generated in windows temporary folder.

**Comment 3**

Also, you do not provide the data used for your study. The manuscript describes the primary generic datasets from where you take data. You must provide detail on the exact input files that you have used from them and, if possible, upload such files to one of the repositories that we can accept according to our data policy (see above). I mean, specifically, the derived DEM data, solar radiation data, meteorological data, and land use and land cover.

**Response**

Thank you for your suggestions. According to your advice, we have provided the input files for both RS and multi-source data driven CASA model in the revised supplement.

**Changes in Code and Data**

(1) The input data for RS data driven CASA model were provided in the revised supplement.

These data were put in the Inputdata folder which contains the subfolder of LUCC (Land-use and Land-cover change), MOD08\_M3, MOD09A1, MOD13Q1, MOD11A2, the files of DEM.tif and study\_area.shp. The subfolder of MOD08\_M3, MOD09A1, MOD13Q1 and MOD11A2 contain the files of cloud cover, band6, band7, NDVI, land surface temperature, which is extracted from the dataset of MOD08\_M3, MOD09A1, MOD13Q1 and MOD11A2 product.

In addition, the dataset of MOD08\_M3, MOD09A1, MOD13Q1 and MOD11A2 product consist of several sub datasets, which are too large (its size is about 1.85 GB) to be include in supplement for unloading, so the codes to extract the sub datasets of cloud cover, band6, band7, NDVI, land surface temperature from these dataset were also provided in the revised supplement.

(2)The solar radiation data, meteorological data and its derived data for calculating NPP with multi source data driven CASA were also provided in the revised supplement. These data are contained in the folder of Multi\_source\_data\_driven\_CASA.

**Comment 4**

Please, be aware that failing to comply with these requirements can result in the rejection of your manuscript. Also, please, reply as soon as possible to this comment with the requested code and data so that it is available for the peer-review process, as it should be.

**Response**

Thank you. According to your advice, we have provided the modified code and the relevant data in the supplement.

Dear Juan A. A ñel, once again, thank you very much for your comments and suggestions.

In case any advice give, please do not hesitate to contact me.

**Changes in Code and Data**

The modified CASA code and its driven data were provided in the revised supplement.

**Response to Referee Comments #1**

**Comments**

Model optimization in NPP estimation is more important for improving model accuracy and model development. The manuscript intended to use remotely sensed data to replace ground observations was a good attempt. However, the introduction and methodology were failed to provide an appropriate design and description, the major concern were:

**Comment 1**

First, as the manuscript planned to MODIS products to replace ground observations data, the authors should summarize the advantages and disadvantages for the replacement of parameters used in CASA model. A comprehensive summary of these parameters from previous researches need to be compared before your chosen. From the view of the present manuscript, the references citations were too limited, and can't offer the reasons why you need to replace the parameters from RS products.

**Response**

Thank you for your suggestions. We have summarized the CASA parameters from previous researches and their advantages and disadvantages in the revised discussion section. We have added some references and modified some inexact statements to offer the further reasons for replacement CASA parameters with RS products.

**Changes in manuscript**

(1) The references (Chen et al., 2017, Fu et al., 2011, Liu et al., 2018, Qie et al., 2020, and Zou et al., 2015) about the development of CASA parameters were added in the revised introduction section.
(2) The CASA parameters from previous researches were summarized and their advantages and disadvantages were analyzed in the revised manuscript (Line 71-74).

"In summary, CASA model is still driven by multi-source data, e.g. RS data and ground observations data. The parameter SOL can be simulated with radar RS data while it should be introduced total cloud cover to improve simulation accuracy. The parameters  $T_{\epsilon 1}$ ,  $T_{\epsilon 2}$  and WSC are dependent on ground meteorological data, soil data and other ground observation points data..."

(3)The reasons for replacing the parameters with RS products were offered in the revised manuscript (Line 81-88).

"Unlike ground observation points data, however, satellite RS can rapidly obtain regional data. Advancements in satellite sensor technologies and RS algorithms have yielded many LUCC data products (e.g., CCI-LC, MCD12, and GlobeLand30) and quality-controlled RS products, which are available online. GlobeLand30, a global LUCC data product, is widely used by scientists and users around the world (Chen et al., 2017). Moderate Resolution Imaging Spectroradiometer (MODIS) satellite sensor records cloud cover and land surface information. Some MODIS products, e.g., land surface temperature (LST) product, were evaluated in several previous studies (Wan et al., 2002; Zou et al., 2015) and applied in terms of air temperature estimation and other fields (Fu et al., 2011; Qie et al., 2020). Therefore, to drive a CASA model with an entire set of RS data, we used the MODIS products, GlobeLand30 product, and DEM data to improve CASA model as follows:...".

**Comment 2**

Second, the manuscript used MODIS products to substitute ground observations. As MODIS product has its own uncertainty, have you evaluate the uncertainty of MODIS used in the study region? As far as I know, some Chinese RS products of SOL, land surface temperature, SWC, and FPAR were generated from the view of parameter localization, comparing with MODIS products. Why not choose these Chinese RS products?

**Response**

**We understand your concern about it.**

Some researchers have evaluated the uncertainty of MODIS products in Qinghai-Tibetan Plateau (QTP). Zou et al. (2015) evaluated the MODIS LST product and found that the RMSE of daytime and nighttime MODIS LST were about 4.41-5.29°C and 3.05-3.78°C respectively. For the QTP, MODIS LST product has been applied to the lower-altitude regions (

Think you for your interesting questions. Qinghai Lake Basin with the area of 29,661 km2 is located in the northeastern QTP. Yet Qinghai Lake Basin is still small relative to the QTP with the area of 2583,700 km2. The word "some" might express more precisely than "small".

**Changes in manuscript**

This imprecise statement was rewritten in the revised manuscript (Line 74-75).

"The spatial distributions of these ground observation points are usually scattered and far apart. In some regions, there may be scant or even no observation stations, which drives down the application of CASA model".

**Comment 10**

In Figure 1, why the samples of NPP field observation was located around the Lake, with no samples in western mountain area. Is this sample representative? A land cover map showed here will be better to demonstrate the grassland distribution of the study region.

**Response**

Thank you for your suggestions. As you know, the western mountain area of Qinghai Lake Basin has a greatly varied terrain and high altitude, which means that a cold climate, high-altitude hypoxia and bad traffic results in difficult sampling. It's even possible to get High Altitude Disease while sampling. This is one reason why we attempt to use RS to drive CASA model. Even so, the samples contained *Kobrecia parva*, *Stipa purpurea*, *Carex pamirensis*, *Achnatherum splendens*, and *Blysmus sinocompressus*, which covered a majority of grassland types in the Qinghai Lake Basin and had a certain representation.

According to your suggestion, a land cover map have been added in the revised Figure 1. Thank you very much for your comments and suggestions.

**Changes in manuscript**

A land cover map (GlobeLand30 product in 2020) was added in the Figure 1 that was redrawn in the revised manuscript.

**Response to Referee Comments #2**

**Comments**

Review of "Improved CASA model based on satellite remote sensing data: Simulating net primary productivity of Qinghai Lake Basin alpine grassland" by Wu et al.

The study "Improved CASA model based on satellite remote sensing data: Simulating net primary productivity of Qinghai Lake Basin alpine grassland" by Wu et al. suggests and tests to drive the CASA model, simulating NPP, with remote sensing data only. To validate the new model formulation, they compare simulated NPP results to alpine grasslands in the Qinghai Lake Basin. Relying only on remote sensing data instead of multi-source data e.g., ground observations has several advantages, therefore the aim of this paper is valid and useful. The paper also demonstrates an improvement in model accuracy for the given study region. I see, however, large problems in the presentation of the methods and results, as well as in the approach to validate the model. Also, the quality of the paper, in general, should be improved to be published in GMD. My major concerns are as follows:

**Comment 1**

There are many problems with the grammar and structure, which sometimes obstruct the understanding of the paper. This issue is prevailing over the whole paper, especially in the introduction.

**Response**

Thank you for pointing this out. We have carefully checked the whole paper and corrected the grammatical and structured errors in the revised manuscript.

**Changes in manuscript**

In the section of Introduction, Data sources, Methods, Results, Discussion and recommendations, and Conclusions, the grammar, structure and spelling mistakes were carefully checked and corrected them accordingly.

**Comment 2**

The authors only use one study region and claim a large improvement in the CASA model. In the Abstract they write to provide a reference for rapidly simulating grassland, farmland, forest, and other vegetation NPP. They also write to satisfy requirements of e.g., precision agriculture. While this is not further discussed in the paper, I would be also careful to make such statements by only comparing to one study region and a limited time window. I would like to have a slightly larger discussion about the study site. In L94 the authors write that it is a typical empirical test site. But since most of the vegetation is grasslands and alpine meadows, other locations should differ a lot. Is there and improve the model for this region. But the authors should be careful in claiming that they achieved general, large-scale model improvement and only provide results for one specific location. Is there a reason, why this site is special and/or important? Would it be difficult to check the new model also for other sites and just compare it to other published NPP values (taking on-site measurements is obviously more complicated)?

**Response**

(1)Thank you for your rigorous comments. According to your nice suggestions, the statements that are not rigorous (e.g., "rapidly simulating grassland, farmland, forest, and other vegetation NPP", "satisfy requirements of precision agriculture", and "a typical empirical study area") have been deleted in the revised manuscript.

(2)Since the field observation NPP data in other sites were not obtained at the time of this study, this study site (Qinghai Lake Basin, its main vegetation types are alpine grasslands and alpine meadows, which account for 85.31% of all vegetation types) was chosen to validate RS driven CASA model. This site is not special. We also hope that the new model (RS driven CASA model) will be validated and applied in other sites.

**Changes in manuscript**

(1)The "rapidly simulating grassland, farmland, forest, and other vegetation NPP" was deleted and rewritten the relevant sentence as "We may provide a workable reference for rapid simulating grassland NPP to satisfy the requirements of accounting carbon stocks and other applications" (Line22-23).

(2)The "application requirements of precision agriculture" was deleted and rewritten this part as "These factors prevent CASA from satisfying the requirements for accounting carbon stocks or other applications" (Line80).

(3)The "a typical empirical study area" was deleted and rewritten the relevant sentence as "Qinghai Lake Basin was taken here as a study area to test the proposed RS data driven CASA model under conditions of varied topography and relative single vegetation types." (Line 99-101).

**Comment 3**

The structure of the paper could be largely improved. For example, the intro should be rewritten to better introduce the state of the art and compare it to what has been done in the study. Sometimes, it is not entirely clear, what has been done in the study and how was CASA used before. Some parts of the Introduction should go to methods (see minor comments), while parts of the Discussion could be in the Introduction. Parts of the results would fit much better to methods (see minor comments). The Discussion generally talks only about very few results of the model and is in large parts more like a summary.

**Response**

(1)Thank you for your valuable suggestions. We agree that the structure of the paper should be partly improved. We have adjusted its structure under the condition of no affecting the reading and scientificity.

(2) According to your nice advice, the temperature stress factor was not talked any longer in the revised discussion section and put it into the revised introduction section.

**Changes in manuscript**

(1) The temperature stress factor was not talked any longer in the revised discussion section. Its contents in the original discussion section were simplified and then put into the revised introduction section (Line 85-87).

"land surface temperature (LST) product, were evaluated in several previous studies (Wan et al., 2002; Zou et al., 2015) and applied in terms of air temperature estimation and other fields"

(2) Other structure changes can be found in the following Response to Comment 9 and Comment 24.

**Comment 4**

I would like a better Discussion about strengths and weaknesses compared to the traditional approach and a better Discussion about the (large) errors between the traditional and new approaches. It would also be good to have an overview of how useful the model could be, especially as in the Abstract the authors write about precision farming, but this is not taken on in the paper. Especially, since the error is up to 50% or 85% (in the traditional approach), I would like to read about how such a large error is possible and how much use the model could have. With such large errors, any improvements should be put into context. For example, how do other models perform? Do they also have such large errors? I also wonder, why some of the RS data has not been used before in the CASA model.

**Response**

We understand your concern about it. The strengths and weaknesses of the traditional approach have been preliminarily discussed in the preprint. According to your nice advice, we further discussed the reason for the large errors in the revised manuscript.

The SOL simulated by traditional approach of sample 7 is 271.39 MJ•m-2 •month-1, which is obtained by interpolating the SOL of observation stations. The average simulated and measured SOL of Gangcha observation station is 434.59 MJ•m-2 •month-1 and 692.71 MJ•m-2 •month-1 respectively (Table 3). The distance of this station from the sample 7 is about 43 km (see the following Fig).

Hence for sample 7, the errors of traditional approach (multi-source data driven CASA) is mainly caused by the parameter SOL and the spatial interpolation method.

Some RS data has been used for calculating several parameters of the CASA model before, for

instance, some studies (e.g., Bao et al., 2016) used RS data to calculate the parameters of WSC, which can be found in the original preprint (L67); we also rewrote some studies that cited in the original preprint, e.g., Wang et al. (2017) used MODIS LUCC product (MCD12Q1) in the CASA model to determine the  $\varepsilon_{max}$  for each vegetation type; in this revised manuscript, we also cited some new studies that used RS data in CASA(e.g., Liu et al., 2018).

**Changes in manuscript**

(1)The reason for causing large errors was added in the revised manuscript (Line 310-314).

"The sample 7 has the maximum errors of estimation NPP (Table 4). Its SOL simulated by traditional approach is 271.39 MJ•m-2 •month-1, which is obtained by interpolating the SOL of observation stations. The average simulated and measured SOL of Gangcha observation station is 434.59 MJ•m-2 •month-1 and 692.71 MJ•m-2 •month-1 respectively (Table 3). The distance of this station from the sample 7 is about 43 km. Hence for sample 7, the errors of multi-source data driven CASA is mainly caused by the parameter SOL and the spatial interpolation method."

(2)Some studies cited in the original preprint were rewritten in this revised manuscript, e.g., "Wang et al. (2017) used MODIS LUCC product (MCD12Q1) in the CASA model to determine the  $\varepsilon_{max}$  for each vegetation type" (Line 44-45).

Some new studies that used RS data in CASA were cited in this revised manuscript, e.g., "Liu et al. (2018) improved WSC by the way of combining RS data and measured soil moisture data." (Line 69).

**Comment 5**

Adapting Table 1 with the different inputs for the "old" and "new" CASA model would greatly help for a better overview".

**Response**

Thanks for your valuable suggestion. We have adopted it and modified the Table 1.

**Changes in manuscript**

The "old" and "new" CASA model were added and revised the title of Table 1 as "Calculation method and input data for CASA model parameters". For the changes in detail, please see the revised Table 1.

**Comment 6**

There are sometimes references missing for several statements or model calculations. If they were developed by the authors, some reasoning or development of the method is missing.

**Response**

Thanks for your nice suggestions. We have added the missing reference (Ministry of Ecology and Environment, PRC, 2021) about calculation dry aboveground biomass and complemented the statements and formulas about calculation SOL in the revised manuscript.

**Changes in manuscript**

(1) The missing reference (Ministry of Ecology and Environment, PRC, 2021) was added in the revised manuscript (Line 136-138).

"Referencing...the technical specification for field observation of grassland ecosystem (Ministry of Ecology and Environment, PRC, 2021), three 1 m  $\times$  1 m quadrats were designed in the corner of square sample plots..."

(2) The formulas of calculation *diffuse\_proportion* and *transmittivity* was added in the revised Table

**3.**

"...*diffuse\_proportion* =0.2+ 0.055*level*, *transmittivity*=0.6-0.055*level*. The step length of 0.055 was determined by repeatedly testing."

**Minor comments:**

**Comment 7**

L31: In many global models, NPP is also calculated and not just an input.

**Response**

Thanks for your suggestion. We have revised this inaccurate sentence.

**Changes in manuscript:**

This inaccurate sentence was revised as "NPP has been the subject of attention from academics and governmental agencies (Wang et al., 2017), which is recognized as a key indicator by the International Biological Program..." in the revised manuscript (Line 29-31).

**Comment 8**

L37/38: Instead of "process models", I would write "process-based models".

**Response**

We are pleased to adopt your nice advice. Thank you.

**Changes in manuscript**

The word "process" was replaced as "process-based" in the revised manuscript (Line 38).

**Comment 9**

L43-L69: This is too much detail for the introduction. This part could be shortened for the relevant details to present the approach, while the details should be in the method section as a model description.

**Response**

Thank you for your valuable advice. Acting on your recommendation, some details have been deleted in the original introduction section. Some details in method or discussion sections have been put into the revised introduction section.

**Changes in manuscript**

(1)The details "The empirical coefficients a and b will change as the time and territories change" was deleted in the revised introduction section.

(2)The part in the original discussion sections "When astronomical solar radiation passes through the atmosphere, it is weakened by atmospheric scattering and absorption, and finally transmits to earth surface (so-called surface solar radiation), which means that atmospheric conditions significantly affect surface solar radiation" was put into the revised introduction section (line 53-55).

**Comment 10**

L44-47: Te1, Te2, and emax sound like plant-specific parameters but the authors write that they are usually calculated by air temperature or RS data. Please clarify.

**Response**

I am sorry that this part was not clear in the original manuscript. The emax is a plant-specific parameters and is cautiously rewritten in the revised manuscript.

Excessively low temperature can limit plant photosynthesis and excessively high temperature can increase the respiration consumption of plants. Temperature stress factors  $T_{\epsilon 1}$  and  $T_{\epsilon 2}$  represent the effects of low temperature and high temperature on Light Use Efficiency of plants.

**Changes in manuscript**

The statement of  $\varepsilon_{max}$  was rewritten in the revised manuscript (line 43-45).

"At regional scales, the FPAR is usually calculated by remote sensing (RS) data (e.g., Potter et al., 1993; Pei et al., 2018), and the  $\varepsilon_{max}$  for vegetation types is usually determined by Land-use and land-cover change (LUCC)"

**Comment 11**

L52-53: How did you determine the coefficients a and b for your time and location? In 4.1.1 you write that they were adopted from Liu et al. But are these values specific for the study region? (But all this should be part of the methods)

**Response**

Thank you for your valuable advice. I am sorry that this part was not clear in the original manuscript. Determining the coefficients a and b need long time serial data sets. The coefficients a (0.24) and b (0.46) were adopted the July values from Liu et al. (2021) for a lack of long serial historic data sets in study region.

**Changes in manuscript**

The values of coefficients a and b were added in the revised Table 1.

"The empirical coefficients a (0.24) and b (0.46) were adopted the July coefficients from Liu et al. (2021)."

**Comment 12**

L81/82: "we hope to use..". The authors should better write what they did and achieved or not achieved.

**Response**

Thank you for your nice advice. According to your recommendation, we have rewritten this statement.

**Changes in manuscript**

The "we hope to use..." was deleted and rewritten this part in the revised manuscript (line 87-88). "Therefore, to drive a CASA model with an entire set of RS data, we used the MODIS products, GlobeLand30 product, and DEM data to improve CASA model and its parameters as follows:..."

**Comment 13**

L87: (5) does not really fit the other (1)-(4), which state the different input variables used. Instead of (5) just write where and to what you apply the model with the new input sources. Also, the sentence "the RS data-driven CASA model was tested with multi-source data-driven CASA model" should be rewritten, because it makes not so much sense as it currently stands.

**Response**

Thank you for your valuable and thoughtful comments. According to on your suggestions, we have rewritten this paragraph.

**Changes in manuscript**

The content of (5) was replaced as " $\epsilon_{max}$  was determined by vegetation types from GlobeLand30

product" in the revised manuscript (Line 91-92).

This part was revised as follows (Line 92-94).

"The improved CASA that is called RS data driven CASA in this paper, was compared with multisource data driven CASA, and was tested with the measured NPP of alpine grassland in Qinghai Lake Basin, in the northeast of QTP, China".

**Comment 14**

L127-132: Is there an example, where this procedure has been done before? Is it a standard procedure to measure AGB? Maybe the authors could provide some literature here.

**Response**

We sincerely appreciate your valuable comments. We have added the following literature in the revised manuscript.

Ministry of Ecology and Environment, PRC.: Technical specification for investigation and assessment of national ecological Status: Field observation of grassland ecosystem, available at :https://www.mee.gov.cn/ywgz/fgbz/bz/bz/bz/bz/bz/bz/b2/02106/W020210615510937790570.pdf, Last modified: 12 May 2021(in Chinese).

**Changes in manuscript**

The relevant literature was added in the revised manuscript (Line 135-138).

"Referencing...the technical specification for field observation of grassland ecosystem (Ministry of Ecology and Environment, PRC, 2021), three 1 m  $\times$  1 m quadrats were designed in the corner of square sample plots ...".

**Comment 15**

L146-152: I don't understand why the factor of 0.5 for the proportion of the radiation which can be absorbed by plants is necessary when FPAR is another input for exactly this. What is the difference between the two factors? And why is 0.5 a constant over all regions and plant types?

**Response**

Thank you. According to Potter et al. (1993), FPAR is the fraction of the incoming photosynthetically active radiation (PAR) intercepted by green vegetation, and the factor of 0.5 accounts for the fact that approximately half of the incoming solar radiation is in the PAR waveband (0.4-0.7 um).

**Changes in manuscript**

A further statement about the factor of 0.5 was made in the revised manuscript (Line 156-157). "0.5 represents the proportion of the radiation which can absorbed by plants (0.4-0.7 um)".

**Comment 16**

L149-152: Here again, the description of Te1, Te2, and emax do not fit the Introduction. Why should e.g., emax be calculated by RS data when it is the maximum possible efficiency? Again, the model description part of the Introduction should be part of 3.1.

**Response**

Thank you for your suggestions. To state the research progress of CASA parameters driven by RS data, it is useful that the parameters Te1, Te2, and emax is described briefly in the Introduction section.

At regional scales, the  $\epsilon_{\text{max}}$  for vegetation types is usually determined by Land-use and land-cover

change (LUCC). Wang et al. (2017) used MODIS LUCC product (MCD12Q1) in the CASA model to determine the  $\varepsilon_{max}$  for each vegetation type.

**Changes in manuscript**

A further instructions for emax was made in the revised manuscript (Line 43-45).

"the  $\varepsilon_{max}$  for vegetation types is usually determined by Land-use and land-cover change (LUCC). Wang et al. (2017) used MODIS LUCC product (MCD12Q1) in the CASA model to determine the  $\varepsilon$ max for each vegetation type".

**Comment 17**

L169-170: Why create 10 levels and not use a continuous result for diffuse\_proportion and transmissivity? How is the linear relationship developed?

**Response**

Thank you for advising this scientific issues. In very clear sky conditions, the typically observed values of transmittivity are 0.6 or 0.7, and the typical values of diffuse\_proportion are 0.2 (https://pro.arcgis.com/en/pro-app/latest/tool-reference/spatial-analyst/area-solar-radiation.htm). So the linear relationship is developed:

diffuse proportion=0.2+0.055 level\_cloud cover

transmittivity=0.6-0.055 levelcloud cover

Under 10 levels of total cloud cover, the step length of 0.055 is determined after repeatedly testing. Under the condition of the continuous total cloud cover ranging from 0 to 10000, it is an interesting and scientific issues for determination the step length.

**Changes in manuscript**

The linear equation of *diffuse\_proportion and transmittivity* was added in the revised manuscript (Line 577-579).

*"diffuse\_proportion* =0.2+ 0.055*level, transmittivity*=0.6-0.055*level.* The step length of 0.055 was determined by repeatedly testing".

**Comment 18**

L173-174: Do you have any citation for the statements in this sentence?

**Response**

We understand your concern about it. The shortwave infrared reflectance is negatively correlated with water content, which is a common point of RS scientific fields. The following literature may partly support this point.

Fensholt, R., and Sandholt, I.: Derivation of a shortwave infrared water stress index from MODIS near and shortwave infrared data in a semiarid environment, Remote Sens Environ., 87, 111-121, doi:10.1016/j.rse.2003.07.002, 2003.

It might be that "relation" is more precise than "RS principle" here.

**Changes in manuscript**

The "RS principle" was replaced as "relation" and the relevant contents was rewritten as follows (Line182-183).

"According to the relation that shortwave infrared reflectance is negatively correlated with surface water content...".

**Comment 19**

*L185-189: Please rewrite this paragraph due to bad English sentence structure. And this paragraph would probably fit better to the methods.*

**Response**

Thank you for pointing this (the SOL simulated by Angstrom-Prescott equation) out. Brief description experimental process here might be helpful to understand the obtaining of SOL results. we have shorten and rewritten this paragraph in the revised manuscript.

**Changes in manuscript**

This paragraph was shorten and rewritten in the revised manuscript (Line195-197).

"The SOL of ground stations were obtained using ground meteorological data and Angstrom-Prescott equation (Table 1). Natural Neighbor spatial interpolation approach was applied to convert the SOL of ground stations into grid SOL over study area (Fig. 2-A)".

**Comment 20**

L203-2004: I would not call the approach superior based on one location. Just write that it yielded better results for the study region.

**Response**

Thank you for your rigorous comments. We have deleted the inexact words such as "superior" and rewritten the relevant statement based on the studied results.

**Changes in manuscript**

The relevant statement was rewritten in the revised manuscript (Line210-211).

"For simulating SOL, the improved approach significantly increased the accuracy in the study area."

**Comment 21**

L207-210: This would also fit better to methods.

**Response**

Thank you again for your suggestion. We have put this (traditional methods of calculation WSC) into the revised Table1 describing the methods and the input data for CASA model parameters.

**Changes in manuscript**

A column to illustrate the method of traditional and model parameters was added in the revised Table1. Additionally, this part was rewritten in the revised manuscript (Line 214-215).

"The WSC of ground stations were obtained using ground meteorological data for July 2020 and approaches listed in Table 1. Natural Neighbour approach was used to convert the WSC of ground stations into grid WSC over study area (Fig. 3-A)."

**Comment 22**

L233-234: Again, you compare the results just to one study area and only to July 2020 but claim a major improvement of the model. For more evidence, it would be beneficial to compare your results to more data. Are there any NPP datasets available for a larger region or a longer period, to which you could easily use and apply the model to?

**Response**

We totally understand your concern. We have cautiously stated the relevant conclusions in the revised manuscript. So far, we do not obtain field observation NPP datasets in other sites. If there are any grassland NPP datasets available for a larger region or a longer period, we are also eagerly to use them to check model!

**Changes in manuscript**

The relevant conclusions were stated cautiously in the revised manuscript (Line 242).

"The simulation results of RS data driven CASA are more in accordance with the measured NPP."

**Comment 23**

L240: Again, I would not write superior, due to scarce evidence, just write that it performed better for the given data points.

**Response**

Thank you for your rigorous comments. We have rewritten this inaccurate statements "RS data driven CASA is superior to the multi-source data driven CASA" as "RS data driven CASA significantly increased the accuracy of grassland NPP in the study area."

**Changes in manuscript**

This inaccurate statements was rewritten in the revised manuscript (Line 243).

"RS data driven CASA significantly increased the accuracy of grassland NPP in the study area."

**Comment 24**

L251-262: Much of this is not really discussed but would probably fit well into the introduction.

**Response**

Thank you for your suggestion. Due to this paper focused specifically on improving the parameters SOL and WSC, if the L251-262(Discussion SOL) were completely put into introduction, it might be difficult to describe the simulation SOL by introducing RS cloud cover. We have shorten this and put the "Astronomical solar radiation passes through the atmosphere, it is weakened by atmospheric scattering and absorption, and finally transmits to earth surface (so called surface solar radiation), which means that atmospheric conditions significantly affect surface solar radiation" into the revised introduction section.

**Changes in manuscript**

Some part of this paragraph was put into the revised introduction section (Line53-55).

"When astronomical solar radiation passes through the atmosphere, it is weakened by atmospheric scattering and absorption, and finally transmits to earth surface (so-called surface solar radiation), which means that atmospheric conditions significantly affect surface solar radiation".

**Comment 25**

L271: What do you mean by that the WSC results of your improved approach are unique?

**Response**

Thank you for pointing this out. I am sorry that this sentence was not clear in the original manuscript. The word "unique" might be replaced by "certain".

This sentence might be written like this "The WSC result of our improved approach is certain as long as the same RS data is input in formula (3)-(5)".

**Changes in manuscript**

This sentence was rewritten in the revised manuscript (Line272).

"The WSC result of our improved approach is certain as long as the same RS data is input in formula (3)-(5)".

**Comment 26**

296-288: Would it not be possible to model NPP for the full year as well? Results could be much easier compared to the reported NPP.

**Response**

Thank you for your insightful suggestions. Of cause, it can be modelled NPP for the full year though monthly NPP of growing season.

Qinghai Lake Basin is located on the Qinghai-Tibetan Plateau, which has a severely cold climate and short growing season. Vegetation is in its growth stage in July and its biomass reaches the highest values for the whole year before the end of August or the beginning of September, which means that grassland NPP also reaches the annual maximum value about a month later.

As you know, field observation data is important for validation model. Because there is no field observation NPP data of other growing season, we just model NPP for the July of 2020 in the paper. In further, once field monthly NPP of other growing season were obtained, we will compare it to the reported annual NPP.

**Changes in manuscript**

A further explanations were made in the revised discussion section (Line 285-289).

"Vegetation is in its growth stage in July and its biomass reaches the highest values for the whole year before the end of August or the beginning of September, which means that grassland NPP also reaches the annual maximum value about a month later. The reported NPP encompasses the full year, so it is reasonable that July NPP simulation values would be lower than some previously reported NPP values."

**Comment 27**

L299: The title of the subsection does not fit the text.

**Response**

Thank you for your insightful suggestion. Because the section 5.4 is about the discussion of simulation results with RS data driven CASA, We have deleted the subsection 5.4.1 and 5.4.2 and rewritten the title of section 5.4.

**Changes in manuscript**

The subsections were deleted and this section was titled as "Rationality of NPP simulation results" (Line 282).

Once again, thank you very much for your comments and suggestions.

**Other changes**

**(1) Changes of coordinate system**

We checked the files in the preprint and found that their coordinate systems were different.

The coordinate systems of the MODIS products, LUCC file (n47\_35\_2020lc030.tif), the DEM file (DEM.tif) and the study area file (study\_area.shp) were Sinusoidal, Universal Transverse Mercator (UTM) grid system, CGCS2000 and CGCS2000 respectively, which caused the coordinate systems of CASA parameters (SOL, FPAR, T2, WSC) inconsistently.

In this revised supplement, the coordinate systems of LUCC file (n47\_35\_2020lc030.tif), the DEM file (DEM.tif) and the study area file (study\_area.shp) were transformed to UTM grid system. The coordinate systems of CASA parameters (SOL, FPAR, T2, and WSC) also were transformed to UTM grid system by revised code.

(2) Changes of CASA parameters

The coordinate system transformation can cause pixel translation, especially, the pixels near the boundary in the study area. It means that the statistical values (mean, maximum and minimum) for pixels of input files\images will change correspondingly, which in turn makes the values of CASA parameters change.

The values of FPAR calculated by inputting the maximum and minimum values of NDVI were changed. The maximum value of FPAR in the preprint and this revised manuscript is 0.95 and 0.94 respectively (see the following Fig).